Plasma-based lipidomics reveals potential diagnostic biomarkers for esophageal squamous cell carcinoma: a retrospective study

Chen Yang 1 2 3
Gu Yixuan 1 2 3
Rong Jinhua 2 3 4
Xu Luyin 1 2 3
Huang Xiancong 2 3
Zhu Jing 2 3
Chen Zhongjian chenzj@zjcc.org.cn 2 3
Mao Weimin maowm@zjcc.org.cn 2 3
1 Department of Medical Oncology, The Second Clinical Medical College of Zhejiang Chinese Medical University , Hangzhou , Zhejiang , China
2 Zhejiang Cancer Hospital, Hangzhou Institute of Medicine (HIM), Chinese Academy of Sciences , Hangzhou , Zhejiang , China
3 Zhejiang Key Laboratory of Diagnosis and Treatment Technology on Thoracic Oncology , Hangzhou , Zhejiang , China
4 College of Pharmaceutical Science, Zhejiang University of Technology , Hangzhou , Zhejiang , China
Sistla Srinivas
Electronic publication date: 2024 Apr 29
Publication date: 2024
Volume: 12
Electronic Location ID: e17272
Received 2023 Nov 24; Accepted 2024 Mar 29
Copyright: ©2024 Chen et al.
Copyright year: 2024
Copyright holder: Chen et al.
License: This is an open access article distributed under the terms of the Creative Commons Attribution License, which permits unrestricted use, distribution, reproduction and adaptation in any medium and for any purpose provided that it is properly attributed. For attribution, the original author(s), title, publication source (PeerJ) and either DOI or URL of the article must be cited.
License URL: https://creativecommons.org/licenses/by/4.0/

Keywords: Esophageal squamous cell carcinoma, Lipidomics, Machine learning, Plasma-based diagnostic model

Funding: National Natural Science Foundation of China 81672315 81302840 Medical and the Health Science Project of Zhejiang Province 2022KY622 2020KY487 Zhejiang Provincial Natural Science Foundation of China LY23H010002 Key R&D Program Projects in Zhejiang Province 2018C04009 This research was supported by the National Natural Science Foundation of China (No. 81672315, 81302840); the Medical and the Health Science Project of Zhejiang Province (2022KY622&2020KY487); the Zhejiang Provincial Natural Science Foundation of China (LY23H010002); and the Key R&D Program Projects in Zhejiang Province (2018C04009). The funders had no role in study design, data collection and analysis, decision to publish, or preparation of the manuscript.

==============================
Background

Esophageal squamous cell carcinoma (ESCC) is highly prevalent and has a high mortality rate. Traditional diagnostic methods, such as imaging examinations and blood tumor marker tests, are not effective in accurately diagnosing ESCC due to their low sensitivity and specificity. Esophageal endoscopic biopsy, which is considered as the gold standard, is not suitable for screening due to its invasiveness and high cost. Therefore, this study aimed to develop a convenient and low-cost diagnostic method for ESCC using plasma-based lipidomics analysis combined with machine learning (ML) algorithms.

Methods

Plasma samples from a total of 40 ESCC patients and 31 healthy controls were used for lipidomics study. Untargeted lipidomics analysis was conducted through liquid chromatography-mass spectrometry (LC-MS) analysis. Differentially expressed lipid features were filtered based on multivariate and univariate analysis, and lipid annotation was performed using MS-DIAL software.

Results

A total of 99 differential lipids were identified, with 15 up-regulated lipids and 84 down-regulated lipids, suggesting their potential as diagnostic targets for ESCC. In the single-lipid plasma-based diagnostic model, nine specific lipids (FA 15:4, FA 27:1, FA 28:7, FA 28:0, FA 36:0, FA 39:0, FA 42:0, FA 44:0, and DG 37:7) exhibited excellent diagnostic performance, with an area under the curve (AUC) exceeding 0.99. Furthermore, multiple lipid-based ML models also demonstrated comparable diagnostic ability for ESCC. These findings indicate plasma lipids as a promising diagnostic approach for ESCC.

Introduction

Esophageal cancer (EC) is the eighth most prevalent malignancy in the world and the sixth leading cause of cancer-related death (Morgan et al., 2022). Histologically, EC can be classified into two distinct subtypes, esophageal squamous cell carcinoma (ESCC) and esophageal adenocarcinoma (EA). The former accounts for 90% of all cases and developing countries bear the burden of 80% of global cases (Liang, Fan & Qiao, 2017). Ongoing research has identified alcohol abuse and smoking as the two most definitive risk factors for ESCC (Reichenbach et al., 2019), and other uncertain risk factors include radiation and pesticide exposure, sedentary lifestyle, and diet with low-fiber intake (Codipilly & Wang, 2022). ESCC progresses rapidly, carries a bleak prognosis, and exhibits a high mortality rate, with a five-year survival rate of less than 20%. Therefore, it is vital to implement better management for ESCC patients.

Early detection and diagnosis are effective strategies to reduce the mortality of ESCC. However, early-stage ESCC often lacks noticeable symptoms, thereby making it prone to being overlooked. The gold standard of diagnosis for ESCC is esophageal endoscopic biopsy (Liang, Fan & Qiao, 2017). Although the endoscopy is highly sensitive, it is accompanied by a considerable financial burden and invasiveness, leading to suboptimal patient compliance and a lack of cost-effectiveness, so it is not suitable for ESCC screening in non-high-risk areas within China (Zhu et al., 2020). Moreover, tumor markers can merely serve as auxiliary diagnosis, such as CEA, CA125 and CA199, which are also elevated within the bloodstream of patients afflicted with other malignancies or inflammation of the digestive tract. Hence, there is an urgent need for accurate, convenient, and less invasive diagnostic methods for ESCC.

Metabolic disorders, including carbohydrate metabolism, amino acid metabolism, nucleotide metabolism, and lipid metabolism, play crucial roles in tumorigenesis (Huang et al., 2020; Kaushik & De Berardinis, 2018; Schmidt et al., 2021). Currently, metabolomics has emerged as a powerful tool for identifying metabolic alterations in various diseases (Li et al., 2021). Lipidomics, as a branch of metabolomics, has gained attraction in cancer research due to the detection of dysregulated lipid metabolism in tumors, including ESCC (Liang et al., 2021; Yuan et al., 2021). However, huge amount of differential lipids detected from lipidomics pose challenges in identifying the most diagnostic one for tumors. While, machine learning (ML) algorithms, known for their data processing capabilities, can effectively select relevant lipid metabolism features and construct diagnostic models for tumors (Ambale-Venkatesh et al., 2017; Kourou et al., 2015; Yuan et al., 2021).

In this study, plasma-based lipidomics was conducted in ESCC patients and healthy controls using liquid chromatography-mass spectrometry (LC-MS) to find potential lipid biomarkers with diagnostic value in ESCC. Further, ML and lipidomics results were combined to develop diagnostic model for early ESCC diagnosis, aiming to explore a novel approach in this area.

Methods & materials

Chemicals and reagents

High performance liquid chromatography (HPLC) grade methanol, acetonitrile (ACN), isopropanol (IPA), methyl-tert-butyl ether (MTBE), and ammonium acetate were purchased from Merck (Darmstadt, Germany). Distilled water was purchased from Wahaha Group Co., Ltd. (Hangzhou, China). Formic acid (HPLC) was purchased from Roe Scientific Inc. (Delaware, USA).

Participants and sample collection

This retrospective study analyzed plasma samples obtained from Zhejiang Cancer Hospital (Hangzhou, China) between December 2010 and December 2012. Plasma samples were collected from 40 pathologically diagnosed ESCC patients and 31 healthy controls (HCs). Table 1 presents the basic clinical information of the participants. Blood was collected from individuals who had fasted overnight and transferred into vials pre-treated with the anticoagulant reagent (ethylenediaminetetraacetic acid disodium potassium salt). Plasma was obtained by centrifuging the blood at 2400xg for 8 min. The samples were then stored at −80 °C until analysis. All procedures involving human participants were conducted in accordance with the ethical standards set by the Ethics Committee of Zhejiang Cancer Hospital (IRB- 2019-66), following the principles of the 1964 Helsinki Declaration and its subsequent amendments or comparable ethical standards. Furthermore, since the samples used in our study came from the biobank of Zhejiang Cancer Hospital, the Medical Ethics Committee of Zhejiang Cancer Hospital waived the need for informed consent.

Table 1 Participant characteristics.

Feature	HCa (n = 31)	ESCCb (n = 40)	P-valuec	
Sex				
Male	26 (83.9%)	32 (80.0%)	NSd	
Female	5 (16.1%)	8 (20.0%)		
Age				
Mean ± SDe	59.4 ± 4.3	60.7 ± 7.8	NS	
BMI				
Mean ± SD	NAf	21.6 ± 2.6	NA	
Stageg				
I	NA	9 (22.5%)	NA	
II	NA	15 (37.5%)		
III	NA	16 (40.0%)		
IV	NA	0 (0.0%)		
Unknown	NA	0 (0.0%)		
Hypertension				
Yes	NA	15 (37.5%)	NA	
No	NA	25 (62.5%)		
Diabetes				
Yes	NA	3 (7.5%)	NA	
No	NA	37 (92.5%)		
Cholesterolemia				
Yes	NA	5 (12.5%)	NA	
No	NA	35 (87.5%)		
Smoking history				
Yes	NA	26 (65.0%)	NA	
No	NA	14 (35.0%)		
Drinking history				
Yes	NA	26 (65.0%)	NA	
No	NA	14 (35.0%)		
Survival status				
Yes	NA	24 (60.0%)	NA	
No	NA	16 (40.0%)		
Notes.

a healthy control.

b patient with esophageal squamous cell carcinoma.

c P-value based on chi-square test or student’s t test, P-value<0.05 represents significant difference between two comparison groups.

d no significance.

e standard deviation.

f not applicable.

g stage of ESCC patients was determined according to pathological TNM classification.

Lipid extraction from plasma

Plasma lipid extraction was conducted referring to previous publication (Yang et al., 2020) with some modifications. Briefly, 300 µL of chilled methanol was added to 40 µL of plasma, followed by mixing for 1 min. Then, 1 mL of MTBE was added and the mixture was shaken at a frequency of 60 Hz for 1 h at room temperature. After that, 250 µL of water was added, and incubated on ice for 10 min. An aliquot of 400 µL of the upper organic phase was transferred to a new tube after separation by centrifugation at 16,200xg at 4 °C for 15 min, and subsequently was dried using a concentrator. The dried residue was reconstituted by mixing with 80 µL ACN-IPA-water (65:30:5, v/v/v) followed by centrifugation at 16,200xg, at 4 °C for 15 min, and an aliquot of 60 µL of supernatant was transferred to sample vial. Finally, 5 µL of supernatant was loaded to liquid chromatography-mass spectrometry (LC-MS) analysis.

The pooled quality control (QC) plasma samples were generated by combining equal aliquots of plasma from each individual sample, which were then dispensed into 40 µL volumes. The extraction process employed for these pooled samples was identical to that used for the individual sample pretreatment.

LC-MS analysis

LC-MS analysis was conducted according to previous research (Yang et al., 2022). In brief, ultimate 3000 UHPLC system coupled with Q exactive orbitrap mass spectrometer (both from Thermo Fisher Scientific, Waltham, MA, USA) was used for lipidomics analysis. Chromatographic separation was carried out on an Acquity UPLC BEH C18 column (2.1 mm × 100 mm, 1.8 µm, Waters, Milford, MA, USA). Solvent A consisted of a mixture of ACN/water (3:2, v/v) containing 0.1% (v/v) formic acid and 10 mM ammonium acetate, while solvent B was composed of IPA/ACN (9:1, v/v) with the same additives. The flow rate was 0.3 mL/min, and column temperature was set at 50 °C. The elution condition was set at 0.0−1.5 min, 32% B; 1.5 min-15.5 min, 32%–85% B; 15.5–15.6 min, 85%–97% B; 15.6–18.0 min, 97% B; 18.0–18.1 min, 97%–32% B; 18.1–20.0 min, 32% B. The settings for the mass spectrometer included a capillary voltage of 3.0 kV and a capillary temperature of 300 °C. The sheath gas flow rate was set to 50 Arb. The auxiliary gas had a flow rate and temperature of 15 Arb and 320 °C, respectively. The scan range was set at m/z 100–1200. The full scan MS had a resolution of 70,000 and an AGC target of 3 × 106. The data-dependent MS/MS had a resolution of 17,500 and an AGC target of 1 × 105. The normalized collision energy was set to 30, 40, and 50 eV, respectively.

The analytical procedure employed a full scan mode to collect data from all samples in the batch. For qualitative quality control (QC) samples, data-dependent acquisition (DDA) mode was utilized. To ensure consistent performance and accuracy during the analysis, QC samples were interspersed within the sample injection sequence. The sequence commenced with three consecutive QC samples, followed by the inclusion of one QC sample every ten samples. The sequence concluded with another three consecutive QC samples. This approach helped to correct for mass spectrometry signal fluctuations and maintain reliable data quality throughout the analysis.

Metabolomics data analysis

Data analysis was performed according to existing articles (Yang et al., 2022) with some modifications. Briefly, the research utilized ProteoWizard’s msconvert tool (https://proteowizard.sourceforge.io/download.html) to convert RAW format data into mzXML format data. The R package xcms was then employed for detecting and extracting ion features, which included tasks such as peak picking and retention time correction. To correct signal shifts, the R package statTarget utilized QC-based random forest signal correction (QC-RFSC). Subsequently, ion feature filtration was conducted. In this step, variables were retained if they had a non-zero value in at least 80% of samples within any single group. However, variables in QC samples with a relative standard deviation (RSD) greater than 30% were excluded. Imputation was performed using the K-nearest neighbors (KNN) algorithm. Prior to chemometrics analysis, the detected ions in each sample belonging to the same class were normalized by setting the sum of their peak areas to 100,000. This rigorous approach ensured that only the most consistent and reliable features were retained for further analysis, thereby significantly improving the overall robustness and reliability of the lipidomics data.

Train set samples were used to discover the differentially expressed plasma lipids between ESCC and HC groups. Firstly, unsupervised principal component analysis (PCA) was employed to visualize the overall separation trend of all samples based on the ion features. Subsequently, a supervised partial least squares discriminant analysis (PLS-DA) was utilized to assess the classification ability of these ion features, yielding a variable importance in projection (VIP) value for each ion feature. Furthermore, the statistical significance of the ion features between the ESCC and HC groups was evaluated using a two-tailed Student’s t-test, and the Benjaminii-Hochberg false discovery rate (FDR) was also calculated. The criteria for defining differentially expressed ion features were as follows: VIP >1.0, adjusted p-value (FDR) <0.05, and fold change (FC) >1.50 or <0.667.

The annotation of lipids was as previously described (Tsugawa et al., 2020). Briefly, the RAW format data was first converted to the abf format using Abf converter (https://www.reifycs.com/abfconverter/). Then, MS-DIAL software ver.4.9 (http://prime.psc.riken.jp/compms/index.html) was used to perform feature detection on all ions with the following parameters: the tolerance of MS1 and MS2 were set 0.01Da and 0.025Da; identification score cut off was set 80%; in positive mode, [M + H]+, [M + NH4]+, [M + Na]+ and [M + H-H2O]+ were selected as the adduct types; in negative mode, [M - H]− was selected as the adduct type; the retention time was tolerance set to 0.05 min and MS1 tolerance was set to 0.015 Da in all ions feature alignment option.

The differential lipids between ESCC and HC groups in both train set and test set were illustrated using heatmap.

Diagnostic significance of plasma lipid

In order to assess the diagnostic value of plasma lipids, receiver operating characteristic (ROC) curve analysis was performed for each differentially expressed lipid in the train set using the R package pROC. This analysis allowed calculation of AUC values. The top nine lipids with the highest AUC values in the ROC curve were selected as the most diagnostic plasma lipids.

To determine the optimal cutoff value in the train set, the threshold was set at the point where the Youden Index (Sensitivity + Specificity − 1) was maximized. This approach aims to find the threshold that maximizes the difference between the true positive rate (sensitivity) and the false positive rate (1 − specificity), striking a balance between sensitivity and specificity.

To evaluate the diagnostic value of the plasma lipids, the cutoff value determined in the train set was used to predict the classification of samples in the test set. The performance of each lipid was then assessed using a confusion matrix, which allowed calculation of diagnostic metrics such as sensitivity, specificity, and accuracy. By following this procedure, the study aimed to identify the most informative plasma lipids for diagnostic purposes and evaluate their performance in classifying samples in both the train and test sets.

In addition to single lipid-based diagnostic models, multiple lipid-based ML models were investigated, including partial least squares (PLS) and random forest (RF) from the caret package, and support vector machine (SVM) from the e1071 package. ROC curves were plotted for each model using the train set, and their prediction performance was evaluated in the test set using confusion matrix calculations.

Relationship between plasma lipid levels and clinical features

To establish the relationship between differentially expressed plasma lipid levels and clinical features such as sex, age, drinking history, smoking history, lymph node metastasis (LNM), and TNM stage, statistical tests were conducted. The Kruskal-Wallis test or Wilcoxon test was employed to compare the lipid levels among different groups in all samples. The p-value of less than 0.05 was considered statistically significant, indicating a significant association between the lipid levels and the clinical features being examined.

Results

Basic characteristics of the participants

This study enrolled 40 patients pathologically diagnosed ESCC and 31 healthy controls (HCs). Age and sex were matched between the two groups without any statistically significant difference, as shown in Table 1. Among the ESCC patients, nine were in stage I, 15 were in stage II, 16 were in stage III while no patients were in stage IV. The samples were randomly divided into a training set (ESCC = 28, HC = 20) and a test set (ESCC = 12, HC = 11).

Metabolic shift between ESCC and HC

Metabolic features were analyzed in train set and depicted in Fig. 1. Following peak picking, retention time alignment, grouping, and signal shift correction, a total of 41,028 ion features were obtained, including 22,591 positive ions and 18,437 negative ions. Based on these metabolic features, PCA and PLS-DA score plots were generated to investigate the differences between ESCC and HC groups. The results exhibited a significant separation between the ESCC and HC groups, indicating notable dysregulation in the plasma lipid profile of ESCC patients (Figs. 1A and 1B). The reliability of the PLS-DA result was further validated using a permutation test (n = 20) (Fig. 1C). Furthermore, based on the criteria of differential lipid (FDR < 0.05; VIP > 1.0; FC > 1.50 or FC < 0.667), a total of 5,899 differential metabolic features were shown in the volcano plot, including 2,654 up-regulated features and 3,245 down-regulated features (Fig. 1D).

Figure 1 Separation of patients with ESCC and healthy controls based on ion features in the train set.

(A) Principal components analysis (PCA) score plot distinguishing ESCC patients from healthy controls based on plasma-detected ions in the train set. C, ESCC patient; N, healthy control. (B) Partial least squares discriminant analysis (PLS-DA) score plot differentiating between ESCC and healthy control groups in the train set. (C) Permutation test (n = 20) confirming the validity of the PLS-DA model. (D) Volcano plot displaying differentially expressed ion features in the train set samples. Red dots representing significantly upregulated ions (VIP > 1.0, FDR < 0.05, fold change > 1.50), blue dots representing significantly downregulated ions (VIP > 1.0, FDR < 0.05, and fold change < 0.667), and grey dots representing ions without significant change.

Differentially expressed plasma lipids in ESCC

A total of 99 differentially expressed plasma lipids were identified and found to exhibit significant differences between individuals from ESCC and HC. Among these lipids, 15 were upregulated (FC >1.5) and 84 were downregulated (FC <0.667) in the plasma of ESCC patients compared to HC. The detailed information of these lipids is summarized in Table 2. Heatmap displaying the expression patterns of the differentially expressed lipids in the train set (Fig. 2A) revealed evident differences between ESCC and HC. Among these lipids, approximately three-fourths exhibited a downward trend in ESCC patients compared to HC. Similar expression patterns were observed in the test set (Fig. S1). The proportion of lipid classifications was presented in the pie chart (Fig. 2B), including 21 fatty acids (FAs), 22 glycerolipids (GLs), 37 glycerophospholipids (GPs), 18 sphingolipids (SPs), and one sphingomyelin (SM). GPs represented the largest proportion of differential lipids at 37%, with 34 downregulated lipids and three upregulated lipids. Among the upregulated differential lipids (Fig. 2C), there were eight FAs, three GPs and SPs, and one GL. Among the downregulated differential lipids (Fig. 2D), there were 13 FAs, 21 GLs, 34 GPs, 15 SPs, and one SM.

Table 2 Dysregulated lipids in ESCC.

Lipid	Mode	Adduct	m/z a	RT b (min)	VIP c	P-value d	FC e	Trend	
DG(37:7)	Pos	[M+H]+	647.46	14.33	1.70	1.58 × 10−16	1.93	Up	
Cer(d18:1/24:1)	Pos	[M+NH4]+	630.62	14.77	1.81	2.08 × 10−5	1.64	Up	
GlcCer(44:4;O3)	Pos	[M+NH4]+	832.66	14.21	1.43	4.60 × 10−6	1.81	Up	
FA(14:0)	Pos	[M+NH4]+	227.20	2.15	1.47	2.71 × 10−8	1.82	Up	
FA(15:4)	Pos	[M+NH4]+	233.16	1.41	1.24	5.37 × 10−6	5.47	Up	
FA(16:1)	Pos	[M+Na]+	253.22	2.32	1.45	1.02 × 10−3	1.94	Up	
FA(18:1)	Pos	[M+NH4]+	281.25	3.66	1.12	1.48 × 10−5	2.08	Up	
FA(19:2)	Pos	[M+NH4]+	293.25	3.22	1.21	3.42 × 10−4	1.83	Up	
FA(19:1)	Pos	[M+Na]+	295.27	4.88	1.16	1.82 × 10−9	2.79	Up	
FA(20:2)	Pos	[M+Na]+	307.27	3.99	2.14	1.12 × 10−4	1.89	Up	
FA(22:2)	Pos	[M+NH4]+	335.30	5.90	1.29	1.01 × 10−4	2.11	Up	
PE(O-18:1_22:4)	Pos	[M+Na]+	778.58	13.56	1.30	1.66 × 10−4	1.55	Up	
PE(O-22:2_20:4)	Pos	[M+NH4]+	804.60	13.73	1.30	3.17 × 10−5	1.68	Up	
PE(O-24:2_20:4)	Pos	[M+NH4]+	832.63	14.45	1.09	4.62 × 10−5	1.73	Up	
GlcCer(d18:2/24:0)	Pos	[M+NH4]+	868.70	14.20	1.42	2.00 × 10−5	1.77	Up	
CAR(18:2)	Pos	[M+NH4]+	424.34	1.77	1.19	5.26 × 10−6	0.48	Down	
DG(14:0_18:2)	Pos	[M+NH4]+	582.51	12.15	1.48	1.84 × 10−6	0.30	Down	
DG(16:0_16:1)	Pos	[M+NH4]+	584.52	12.96	1.51	2.77 × 10−4	0.45	Down	
DG(16:0_18:2)	Pos	[M+NH4]+	610.54	13.10	1.39	1.03 × 10−4	0.55	Down	
DG(16:0_18:1)	Pos	[M+Na]+	612.56	13.82	1.41	9.76 × 10−4	0.65	Down	
DG(34:2)	Pos	[M+Na]+	615.50	13.07	1.59	1.05 × 10−4	0.61	Down	
DG(18:1_18:2)	Pos	[M+H]+	636.56	13.13	1.57	3.81 × 10−3	0.66	Down	
DG(18:0_18:1)	Pos	[M+H]+	640.59	14.58	1.27	2.47 × 10−3	0.57	Down	
DG(36:1)	Pos	[M+Na]+	645.54	14.57	1.45	3.25 × 10−3	0.60	Down	
DG(20:1_18:2)	Pos	[M+H]+	664.59	13.91	1.02	7.53 × 10−4	0.60	Down	
DG(38:3)	Pos	[M+H]+	669.54	13.91	1.12	5.57 × 10−4	0.63	Down	
TG(12:0_16:1_18:1)	Pos	[M+H]+	792.71	16.46	1.51	1.68 × 10−3	0.30	Down	
TG(12:0_16:0_18:1)	Pos	[M+H]+	794.72	16.90	1.31	8.34 × 10−3	0.33	Down	
TG(12:0_18:1_18:2)	Pos	[M+H]+	818.72	16.46	1.21	3.67 × 10−4	0.34	Down	
TG(14:0_16:0_18:2)	Pos	[M+H]+	820.74	16.92	1.38	2.67 × 10−3	0.51	Down	
TG(14:0_18:2_18:2)	Pos	[M+H]+	844.74	16.51	1.41	1.27 × 10−4	0.49	Down	
TG(15:1_18:1_18:2)	Pos	[M+H]+	858.75	16.75	1.33	1.03 × 10−4	0.53	Down	
TG(16:0_18:2_18:3)	Pos	[M+NH4]+	870.75	16.60	1.33	3.32 × 10−4	0.52	Down	
TG(16:0_16:1_20:5)	Pos	[M+H-H2O]+	873.69	16.12	1.34	1.68 × 10−4	0.59	Down	
TG(17:1_18:2_18:2)	Pos	[M+H]+	889.73	16.75	1.41	3.08 × 10−5	0.59	Down	
LPC(14:0)	Pos	[M+H]+	468.31	1.62	1.61	3.05 × 10−5	0.47	Down	
LPC(O-16:0)	Pos	[M+H]+	482.36	2.86	1.61	1.32 × 10−3	0.66	Down	
LPC(15:0)	Pos	[M+H]+	504.31	1.92	1.43	8.37 × 10−5	0.56	Down	
LPC(18:3)	Pos	[M+H]+	518.32	1.53	1.54	1.29 × 10−2	0.62	Down	
LPC(18:2)	Pos	[M+Na]+	520.34	1.88	1.23	7.07 × 10−3	0.64	Down	
LPC(18:0)	Pos	[M+H]+	524.37	3.74	1.61	5.98 × 10−5	0.61	Down	
LPC(19:0)	Pos	[M+H]+	538.39	4.67	1.14	6.31 × 10−4	0.63	Down	
LPC(22:0)	Pos	[M+H]+	580.43	7.63	1.98	1.99 × 10−3	0.66	Down	
PC(O-36:5)	Pos	[M+H]+	766.57	12.15	1.61	5.95 × 10−5	0.51	Down	
PC(O-16:1_22:6)	Pos	[M+H]+	790.57	11.40	1.47	8.91 × 10−5	0.65	Down	
PC(O-17:1_21:4)	Pos	[M+H-H2O]+	794.60	13.10	1.44	5.73 × 10−4	0.58	Down	
PI(38:3)	Pos	[M+H]+	906.61	11.11	1.22	2.28 × 10−4	0.63	Down	
Cer(d18:2/23:0)	Pos	[M+NH4]+	634.61	14.52	1.13	8.23 × 10−5	0.63	Down	
SM(30:1;O2)	Pos	[M+NH4]+	647.51	8.21	1.21	4.52 × 10−6	0.50	Down	
SM(31:1;O2)	Neg	[M-H]−	661.53	9.04	1.50	7.99 × 10−6	0.52	Down	
SM(32:2;O2)	Neg	[M-H]−	673.53	8.44	1.68	1.03 × 10−4	0.65	Down	
SM(32:1;O2)	Neg	[M-H]−	675.54	9.82	1.44	1.25 × 10−5	0.66	Down	
Cer(d24:1/19:0)	Neg	[M-H]−	686.64	15.68	1.08	1.94 × 10−3	0.54	Down	
SM(33:2;O2)	Neg	[M-H]−	687.54	9.30	2.08	1.93 × 10−5	0.60	Down	
SM(36:3;O2)	Neg	[M-H]−	727.57	10.32	1.38	3.81 × 10−3	0.55	Down	
SM(39:1;O2)	Neg	[M-H]−	773.65	13.70	1.16	8.58 × 10−10	0.54	Down	
SM(d18:1/23:0)	Neg	[M-H]−	801.68	14.48	1.78	1.48 × 10−5	0.63	Down	
SM(43:1;O2)	Neg	[M-H]−	829.72	15.05	1.25	8.33 × 10−5	0.45	Down	
SM(d18:1/30:6)	Neg	[M-H]−	887.71	16.36	2.00	1.54 × 10−3	0.58	Down	
DG(16:1_18:2)	Neg	[M-H]−	608.52	12.21	1.23	4.99 × 10−3	0.60	Down	
DG(18:2_18:2)	Neg	[M-H]−	634.54	12.39	2.11	2.97 × 10−3	0.51	Down	
FA(13:1)	Neg	[M-H]−	211.17	1.43	2.20	8.49 × 10−5	0.61	Down	
FA(16:0;O)	Neg	[M-H]−	271.23	1.11	2.44	1.79 × 10−10	0.60	Down	
FA(21:1)	Neg	[M-H]−	323.30	6.30	1.78	6.05 × 10−8	0.46	Down	
FA(25:0)	Neg	[M-H]−	381.38	11.41	2.14	8.98 × 10−11	0.42	Down	
FA(27:1)	Neg	[M-H]−	407.39	10.54	2.24	1.32 × 10−10	0.36	Down	
FA(28:7)	Neg	[M-H]−	409.31	8.12	2.37	1.42 × 10−16	0.22	Down	
FA(27:0)	Neg	[M-H]−	409.41	12.62	1.21	9.92 × 10−7	0.57	Down	
FA(28:0)	Neg	[M-H]−	423.42	11.05	2.41	1.19 × 10−8	0.23	Down	
FA(36:0)	Neg	[M-H]−	535.55	14.56	2.41	2.85 × 10−10	0.19	Down	
FA(39:0)	Neg	[M+CH3COO]−	577.60	15.44	1.60	2.90 × 10−13	0.13	Down	
LPI(18:2)	Neg	[M+CH3COO]−	595.29	1.35	1.24	2.02 × 10−3	0.64	Down	
FA(42:0)	Neg	[M-H]−	619.65	16.17	1.52	4.36 × 10−15	0.14	Down	
FA(44:0)	Neg	[M-H]−	647.68	16.72	1.46	1.26 × 10−14	0.11	Down	
Cer(d16:1/23:0)	Neg	[M-H]−	666.61	14.49	1.29	3.54 × 10−6	0.61	Down	
Cer(d16:1/24:1)	Neg	[M+CH3COO]−	678.61	14.15	1.31	1.70 × 10−3	0.64	Down	
PE(O-16:1_18:2)	Neg	[M-H]−	698.52	12.21	1.11	1.01 × 10−4	0.51	Down	
PE(O-16:0_18:1)	Neg	[M-H]−	702.55	13.13	1.36	6.29 × 10−5	0.59	Down	
PE(O-17:1_18:2)	Neg	[M+CH3COO]−	712.54	12.70	1.42	1.33 × 10−3	0.51	Down	
Cer(d19:1/24:0)	Neg	[M-H]−	722.67	15.69	1.64	7.24 × 10−4	0.45	Down	
PE(O-18:2_18:2)	Neg	[M-H]−	724.54	12.24	1.60	6.76 × 10−3	0.66	Down	
PE(O-18:1_18:2)	Neg	[M-H]−	726.55	13.16	1.10	5.70 × 10−4	0.58	Down	
SM(d18:2/14:0)	Neg	[M-H]−	731.54	8.45	1.01	1.85 × 10−4	0.65	Down	
PE(O-17:1_20:4)	Neg	[M-H]−	736.54	12.50	1.26	2.18 × 10−5	0.40	Down	
PE(O-16:1_22:6)	Neg	[M-H]−	746.52	11.65	1.07	7.17 × 10−6	0.58	Down	
PE(O-18:0_20:4)	Neg	[M-H]−	752.57	13.13	1.39	4.67 × 10−3	0.50	Down	
PE(O-18:1_20:3)	Neg	[M-H]−	752.57	13.35	1.46	1.37 × 10−2	0.66	Down	
PE(O-17:1_22:6)	Neg	[M-H]−	760.54	12.16	1.34	5.70 × 10−4	0.60	Down	
PE(O-19:1_20:4)	Neg	[M-2H]2−	764.57	13.38	1.33	8.85 × 10−3	0.52	Down	
PE(O-18:3_22:6)	Neg	[M-2H]2−	770.52	10.82	1.51	1.03 × 10−4	0.59	Down	
PE(O-18:1_22:6)	Neg	[M-H]−	774.55	12.65	1.39	1.22 × 10−4	0.59	Down	
CL(26:0_18:2_18:2_18:2)	Neg	[M-H]−	781.56	11.03	1.36	3.78 × 10−4	0.49	Down	
CL(28:0_17:2_18:2_18:2)	Neg	[M-H]−	788.55	10.33	1.16	1.19 × 10−4	0.49	Down	
PI(16:0_18:2)	Neg	[M-H]−	833.53	9.73	1.35	2.63 × 10−3	0.66	Down	
PE(22:4_22:5;O)	Neg	[M-H]−	856.56	12.65	1.03	2.64 × 10−4	0.66	Down	
PI(16:0_20:3)	Neg	[M-H]−	859.54	10.02	1.34	5.54 × 10−3	0.62	Down	
PI(18:1_18:2)	Neg	[M-H]−	859.54	9.82	1.84	1.22 × 10−4	0.62	Down	
PI(18:0_18:1)	Neg	[M+CH3COO]−	863.57	11.62	1.40	5.68 × 10−8	0.54	Down	
PI(18:0_20:3)	Neg	[M-H]−	887.57	11.12	1.30	4.56 × 10−4	0.64	Down	
Notes.

a Mass to charge ratio.

b Retention time.

c Variable importance in projection.

d P-value based on student’s t test, P-value <0.05 represents significant difference between two comparison groups.

e Fold change.

Figure 2 Differential plasma lipids detected in the train set.

(A) Heatmap clustering the 99 differential lipids in train set revealed a lipid metabolism shift in ESCC patients (n = 28) compared with healthy controls (n = 20). (B) Pie chart showing the proportion of lipid classifications among all the differential lipids above. (C) Pie chart showing the proportion of lipid classifications among up-regulated differential lipids above. (D) Pie chart showing the proportion of lipid classifications among down-regulated differential lipids above. FA, fatty acid; GL, glycerolipid; GP, glycerophospholipid; SM, sphingomyelin; SP, sphingolipid.

Diagnostic performance of the differential plasma lipids

ROC curve analysis revealed that plasma lipids show promise as diagnostic biomarkers for ESCC. The top nine lipids with the highest AUC values were FA 15:4, FA 27:1, FA 28:7, FA 28:0, FA 36:0, FA 39:0, FA 42:0, FA 44:0, and DG 37:7. Eight of these nine lipids achieved an AUC value of 1.00, indicating excellent diagnostic accuracy. FA 15:4 and DG 37:7 showed an up-regulated trend in ESCC samples, while the remaining lipids showed a down-regulated trend (Fig. 3).

Figure 3 Receiver operating characteristic (ROC) and boxplot of the top nine diagnostics lipids.

Receiver operating characteristic (ROC) curves of single-lipid predictive models and boxplots of peak intensity distribution. (A) FA 15:4, (B) FA 27:1, (C) FA 28:7, (D) FA 28:0, (E) FA 36:0, (F) FA 39:0, (G) FA 42:0, (H) FA 44:0, (I) DG 37:7. AUC, area under the curve. P, P-value. FC, fold change.

In testing step, the confusion matrix charts (Fig. 4) were constructed for a more intuitive representation of the diagnostic performance of the top nine lipids. All the nine lipids depicted in the chart exhibited prediction accuracy exceeding 85%, with six exceeding 95%. And seven of these nine lipids achieving 100% diagnostic efficiency in detecting tumors. The top 15 lipids, ranked by their prediction accuracy in the test set, were selected. Table 3 presents their AUC values in the train set, as well as prediction accuracy, sensitivity, specificity, precision, and recall values in the test set. Among these lipids, 13 belonged to FAs, and 11 lipids (FA 27:1, FA 28:7, FA 28:0, FA 39:0, FA 42:0, FA 44:0, FA 22:2, FA 36:0, FA 25:0, FA 19:1, FA 27:0) achieved prediction accuracy exceeding 0.90. The results for the remaining differential lipids can be found in the Table S1.

Figure 4 Diagnostic performance of the single lipid-based diagnostic models in the test set.

Confusion matrix table constructed using confusion matrix algorithms in the test set to show the diagnostic accuracy of the previously nine differential lipids above. (A) FA 15:4, (B) FA 27:1, (C) FA 28:7, (D) FA 28:0, (E) FA 36:0, (F) FA 39:0, (G) FA 42:0, (H) FA 44:0, (I) DG 37:7.

Multiple lipid-based models were constructed using the training set data. The ROC curves in Figs. 5A, 5B and 5C showed high AUC values of 0.990, 0.990, and 0.980, indicating excellent prediction performance. These results were consistent with the performance of individual differential lipids. The accuracy of the models was further validated in the test set, achieving accuracies of 95.7% (Figs. 5D, 5E and 5F). These findings demonstrate the effectiveness of the multiple lipid-based models as accurate diagnostic tools for ESCC.

Relationship between plasma lipid levels and clinical features

The results revealed that 14 differentially expressed plasma lipids were associated with age (Fig. S2), 14 ones with sex (Fig. S3), 18 ones with smoking history (Fig. S4), and only one with drinking history (Fig. S5).

Compared the ESCC patients with and without lymph node metastasis, there were 11 differential lipids showed significant difference (P-value <0.05), including FA 14:0, FA 15:4, FA 27:1, FA 28:7, FA 27:0, FA 42:0, DG 37:7, PC O-16:1_22:6, PC O-17:1_21:4, SM d18:1/23:0, PI 18:0_20:3. Among them, six were up regulation and five were down regulation in ESCC patients with lymph node metastasis. Boxplots illustrating the expression levels of these 11 lipids were shown in Fig. 6. Out of the 99 differential lipids, 10 lipids—FA 14:0, FA 15:4, FA 28:7, FA 27:0, FA 28:0, DG 37:7, PE O-17:1_22:6, PE O-18:3_22:6, PC O-16:1_22:6, and SM d18:1/23:0—exhibited significant differences among the three stage groups (P-value <0.05). Boxplots depicting the expression levels of these 10 lipids are shown in Fig. 7. Among them, FA 15:4 displayed the most prominent upregulated trend in stage III compared to stages I and II, with the P-value of 7.27e−07. Only one of these 10 lipids, PE O-18:3_22:6, exhibited statistically significant differences in pairwise comparisons across stages I, II, and III, with an increase in expression levels in stage II but declining in stage III patients.

Table 3 Lipids of top 15 accuracy in prediction of test set by ROC analysis.

	Train Set	Test Set	
Lipid	AUC a	Accuracy	Sensitivity	Specificity	Precision b	Recall c	
FA 27:1	1.000	0.957	1.000	0.909	0.923	1.000	
FA 28:7	1.000	0.957	1.000	0.909	0.923	1.000	
FA 28:0	1.000	0.957	1.000	0.909	0.923	1.000	
FA 39:0	1.000	0.957	1.000	0.909	0.923	1.000	
FA 42:0	1.000	0.957	1.000	0.909	0.923	1.000	
FA 44:0	1.000	0.957	1.000	0.909	0.923	1.000	
FA 22:2	0.830	0.957	0.917	1.000	1.000	0.917	
FA 36:0	1.000	0.913	1.000	0.818	0.857	1.000	
FA 25:0	0.990	0.913	0.917	0.909	0.917	0.917	
FA 19:1	0.940	0.913	0.833	1.000	1.000	0.833	
FA 27:0	0.920	0.913	0.917	0.909	0.917	0.917	
FA 15:4	1.000	0.870	0.750	1.000	1.000	0.750	
DG 37:7	0.990	0.870	0.833	0.909	0.909	0.833	
SM 39:1;O2	0.980	0.870	0.917	0.818	0.846	0.917	
FA 21:1	0.930	0.870	1.000	0.727	0.800	1.000	
Notes.

a Area under the curve of ROC.

b Positive predictive value.

c True positive rate.

Figure 5 Receiver operating characteristic (ROC) and diagnostic performance of the multiple lipid-based diagnostic models in the test set.

Receiver operating characteristic (ROC) curves of multiple-lipids predictive models constructed using PLS (A), RF (B), and SVM (C) algorithms in the train set. Confusion matrix tables constructed using confusion matrix algorithms in the test set to assess the diagnostic accuracy of the multiple-lipids predictive model above, (D) PLS, (E) RF, and (F) SVM.

Figure 6 Relationship between lymph node metastasis and differentially expressed plasma lipids.

Boxplots illustrating the distribution of peak relative intensity of differential lipid in patients with or without lymph node metastasis. YES: ESCC patients with lymph node metastasis, NO, ESCC patients without lymph node metastasis. Wilcoxon test [1] was performed to identify differential lipids associated with lymph node metastasis in ESCC patients. Lipids were selected based on a significance threshold of P-value < 0.05. * P ≤ 0.05, ** P ≤ 0.01, and **** P ≤ 0.0001.

Figure 7 Relationship between tumor stage and differential expressed lipids.

Boxplots illustrating the distribution of peak relative intensity of differential lipid in ESCC patients across I, II, and III stages. Kruskal-Wallis test was performed to identify differential lipids associated with TNM stage in ESCC patients. Lipids were selected based on a significance threshold of P-value < 0.05. * P ≤ 0.05, ** P ≤ 0.01, *** P ≤ 0.001, and **** P ≤ 0.0001.

Discussion

ESCC is characterized by rapid progression and poor prognosis, therefore, early detection and screening are particularly important (Napier, Scheerer & Misra, 2014). Compared to current traditional methods like esophageal endoscopy biopsy and imaging examination, plasma analysis conducted in our study offers the advantages of non-invasiveness and convenience, regardless of equipment and operator expertise (Codipilly et al., 2018; He & Ke, 2020). Meanwhile, the increasing attention on blood tests for disease surveillance is attributed to their specific ability to reveal alterations within the internal environment. While the sensitivity and specificity of blood tests for ESCC screening, involving auto-antibodies to tumor-associated antigens (TAAs), circulating tumor cells (CTCs), and circulating miRNA, can significantly vary among patients, pathological types, and disease stages (Codipilly et al., 2018; He & Ke, 2020). To achieve a non-invasive, convenient, and highly accurate diagnostic method for ESCC, our study utilized plasma-based lipidomics analysis combined with ML techniques to develop an early diagnostic model based on plasma lipids. The model was constructed using PLS, PF, and SVM algorithms and validated using a test set, thereby enhancing the reliability of the proposed approach.

This study provides compelling evidence of significant dysregulation in the plasma lipid profile of patients with ESCC compared to HC. Dysregulated lipids in ESCC patients include fatty acids (FA), diacylglycerols (DG), and triglycerides (TG). FA exhibit both upregulation and downregulation in ESCC patients. Upregulation of FA in the plasma can be attributed to enhanced FA synthesis capacity in tumor cells, facilitated by abnormal expression and activity of FA synthesis-related enzymes such as fatty acid synthase (FASN), ATP-citrate lyase (ACLY), stearoyl-CoA desaturase (SCD1), and acetyl-CoA carboxylase (Lien et al., 2021). Transcriptional regulation by key transcription factors like SREBP1 and PRP19 (Zhang et al., 2023), as well as activation of signaling pathways such as PI3K/Akt/mTOR and MAPK pathways (Yu et al., 2022), may also contribute to FA upregulation and enhance tumor cell proliferation. However, tumor cells may increase FA uptake and utilization, which can lead to a decrease in FA concentration. The heightened metabolic activity of tumor cells promotes FA oxidation for energy production, biological membrane synthesis, and signal transduction (Yuan et al., 2021), further contributing to lower FA concentrations in the plasma. DG and TG consistently show a downward trend in ESCC patients. The downregulated concentrations of DG and TG in the plasma may be linked to the heightened metabolic activity of tumor cells. Rapidly proliferating tumor cells have increased energy demands, resulting in the consumption of DG and TG through enhanced FA oxidation and energy utilization. This phenomenon explains the observed downregulation of DG and TG levels in the plasma in this study.

In terms of diagnostic models, traditional single biomarker models were basically used in previous research. However, with the increasing advancements in omics research, there has been a rise in the use of ML-assisted multivariate models. In this context, it is worth discussing that our study demonstrated excellent diagnostic performance for individual lipids, while the ML-based multiple-lipids models did not exhibit significantly better diagnostic performance. There are a few reasons accountable for this phenomenon. Firstly, single-lipid models exhibit excellent diagnostic performance, rendering multiple-lipids model with no significant improvement in diagnostic accuracy. Secondly, during plasma sample collection, we overlooked factors such as the disease progression and different therapeutic interventions, potentially resulting in multicollinearity and overfitting during data modeling, thereby impacting the diagnostic performance of the multiple-lipids model. Lastly, a small sample size, indeed, hampers the effectiveness of multivariate modeling, limiting the ability to highlight its advantages. This issue can be addressed by expanding the sample size, as it allows for a more comprehensive analysis of multiple factors. In our future research, we will make additional efforts in data collection and group refining to optimize the diagnostic models.

Several studies have shown that lipid metabolism may have an impact on tumor progression. Jiao et al.’s (2023) study revealed the influence of lipid metabolism-related enzyme LPCAT on the progression of ESCC. Therefore, the Kruskal–Wallis test and Wilcoxon test were performed to reveal the combination of differential lipids with clinical information of ESCC patients. We observed distinct lipid metabolism profiles between samples with or without lymph node metastasis. Furthermore, the metabolic levels of the same lipid varied across different stages of ESCC. Our findings suggest that lipid metabolism levels may serve as indicative factors for patient staging and the presence of lymph node metastasis.The present study has limitations that should be acknowledged. The small sample size from a single center restricts the robustness and generalizability of the data, as it may lack representativeness, be prone to bias, have reduced statistical power, and limited applicability to other populations or settings. Additionally, the absence of a control group comprising high-risk individuals with benign lesions is a limitation. Including such a control group would enable the evaluation of the diagnostic model’s accuracy, sensitivity, and specificity for ESCC. Moreover, it would facilitate the identification of specific metabolic profiles that differentiate ESCC from benign lesions, assess test performance, and enhance internal validity. Furthermore, the study only utilized plasma samples for lipid metabolism profiling analysis, without considering serum or tissue samples. To improve future studies, it is recommended to increase the sample size, involve multiple centers, include a control group with benign lesions, and incorporate a wider range of experimental samples from serum and tissue. These enhancements would contribute to more robust and comprehensive research.

Conclusion

In conclusion, this study has developed a novel and valuable diagnostic model for ESCC by integrating plasma-based lipidomics and ML algorithms, enabling more efficient and accurate clinical diagnosis. Furthermore, the identified prognostic lipid markers exposed the dysregulated lipid metabolism in ESCC, which may provide new therapeutic targets to guide clinical treatment. In summary, this study has improved the understanding in the field of cancer diagnostic model construction by combining metabolomics and ML algorithms. This approach holds promise for cancer diagnosis and has the potential to promote the cancer treatment. However, it is important to acknowledge that the current model only focuses on ESCC, and separate models for different pathological types of esophageal cancer need to be developed and validated. Moreover, the heterogeneity of metabolic characteristics among different patients and disease stages should be considered, and further optimization of the diagnostic model is necessary. There is still substantial work to be done before the model can be effectively implemented in clinical settings, highlighting the need for ongoing research in this area.

Supplemental Information

Figure S1 Heatmap clustering the 99 differential lipids in test set revealed a lipid metabolism shift in ESCC patients (n = 12) compared with healthy controls (n = 11)

Figure S2 Boxplots illustrating the distribution of peak relative intensity of differential lipid related to factor of age in ESCC patients

0: the age of ESCC patients <60 years, 1: the age of ESCC patients ≥60 years. Wilcoxon test was performed to identify differential lipids associated with the factor of age in ESCC patients. Lipids were selected based on a significance threshold of P-value < 0.05.

Figure S3 Boxplots illustrating the distribution of peak relative intensity of differential lipid related to factor of sex in ESCC patients

Wilcoxon test was performed to identify differential lipids associated with the factor of sex in ESCC patients. Lipids were selected based on a significance threshold of P-value < 0.05.

Figure S4 Boxplots illustrating the distribution of peak relative intensity of differential lipid related to factor of smoking in ESCC patients

Wilcoxon test was performed to identify differential lipids associated with the factor of smoking in ESCC patients. 0: ESCC patients without smoking history, 1: ESCC patients with smoking history. Lipids were selected based on a significance threshold of P-value < 0.05.

Figure S5 Boxplots illustrating the distribution of peak relative intensity of differential lipid related to factor of drinking in ESCC patients

Wilcoxon test was performed to identify differential lipids associated with the factor of drinking in ESCC patients. 0: ESCC patients without drinking history, 1: ESCC patients with drinking history. Lipids were selected based on a significance threshold of P-value < 0.05.

Table S1 Prediction results of differential lipids in the test set

Table S2 Survival analysis results of the 99 differential lipids in train set using R package of “survival ” and “survminer ”

Table S3 Raw data of clinical information which contains information of 40 patients with ESCC and 31 healthy controls

Supplemental Information 9 Codes for metabolomics raw data preprocessing

Supplemental Information 10 Positive ions after correction

Supplemental Information 11 All negative ions before they are corrected

Supplemental Information 12 Negative ions are annotated and converted into lipids

Supplemental Information 13 Positive ions are annotated and converted into lipids

Supplemental Information 14 STROBE Statement

Checklist of items that should be included in reports of case-control studies.

We thank Biobank in Zhejiang Cancer Hospital for providing all the samples in the study.

Additional Information and Declarations

Competing Interests

Author Contributions

Human Ethics

Data Availability

The authors declare there are no competing interests.

Yang Chen conceived and designed the experiments, performed the experiments, analyzed the data, prepared figures and/or tables, authored or reviewed drafts of the article, and approved the final draft.

Yixuan Gu conceived and designed the experiments, performed the experiments, analyzed the data, prepared figures and/or tables, authored or reviewed drafts of the article, and approved the final draft.

Jinhua Rong conceived and designed the experiments, performed the experiments, analyzed the data, prepared figures and/or tables, and approved the final draft.

Luyin Xu conceived and designed the experiments, performed the experiments, analyzed the data, prepared figures and/or tables, authored or reviewed drafts of the article, and approved the final draft.

Xiancong Huang conceived and designed the experiments, performed the experiments, analyzed the data, authored or reviewed drafts of the article, and approved the final draft.

Jing Zhu conceived and designed the experiments, performed the experiments, analyzed the data, authored or reviewed drafts of the article, and approved the final draft.

Zhongjian Chen conceived and designed the experiments, performed the experiments, analyzed the data, prepared figures and/or tables, authored or reviewed drafts of the article, and approved the final draft.

Weimin Mao conceived and designed the experiments, performed the experiments, analyzed the data, authored or reviewed drafts of the article, supervision, and approved the final draft.

The following information was supplied relating to ethical approvals (i.e., approving body and any reference numbers):

Medical Ethics Committee of Zhejiang Cancer Hospital (IRB - 2019-66).

The following information was supplied regarding data availability:

The raw data are available in the Supplementary Files.

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
