# Peer review of "Plasma-based lipidomics reveals potential diagnostic biomarkers for esophageal squamous cell carcinoma: a retrospective study"

_PeerJ, doi:10.7717/peerj.17272_

## Round 0.1 · original submission · Major Revisions

Please attend to the comments by the three reviewers and provide a detailed explanation of work revised or rebuttal. Also please check the manuscript thoroughly for grammatical errors. I agree with the reviewer that the sample size is small but you can comment on that in the manuscript.

**Language Note:** The review process has identified that the English language must be improved. PeerJ can provide language editing services - please contact us at copyediting@peerj.com for pricing (be sure to provide your manuscript number and title). Alternatively, you should make your own arrangements to improve the language quality and provide details in your response letter. – PeerJ Staff

Reviewer 1 ·

Basic reporting

No comments

Experimental design

No comments

Validity of the findings

No comments

Additional comments

Authors did really excellent job in ESCC analysis filed. However, lipid metabolism is a very complex process, especially for esophageal cancer patients, so I hope the authors can add BMI information of the patients in clinical information table.

Reviewer 2 ·

Basic reporting

No comment

Experimental design

No comment

Validity of the findings

1. Have you ever tried to explore the association between lipidomics/plasma lipids with patients' outcome, e.g., overall survival? If not, please do so.

2. Is it possible to measure the concentration of different plasma lipids?

Reviewer 3 ·

Basic reporting

requested for professional English editing.

Experimental design

The sample size is too small to draw conclusions

Validity of the findings

The manuscript “Plasma-based lipidomics reveals potential diagnostic biomarkers for esophageal squamous cell carcinoma: a retrospective study” provides valuable information about developing a convenient and diagnostic method for esophageal squamous cell carcinoma (ESCC) using plasma-based lipidomics analysis.

Following are some crucial concerns:

1. Could you elaborate on the specific mechanisms by which dysregulated fatty acids (FA), diacylglycerols (DG), and triglycerides (TG) contribute to the plasma lipid profile in ESCC patients? Additionally, are there any known regulatory pathways or molecular processes involved in the observed upregulation and downregulation of FA, DG, and TG?

2. Considering the study’s limitation of a small sample size from a single center, how might this affect the robustness and generalizability of the data?

3. How essential is it to include a control group comprising high-risk individuals with benign lesions, and how might their inclusion impact the study's outcomes?

4. The absence of a control group comprising high-risk individuals with benign lesions. Can you explain how crucial the inclusion of such a control group is concerning enhancing the validity?

5. Could you please provide a table presenting "The clinical and demographic characteristics of all subjects" (smoking, drinking, Hypertension, BMI, cholesterolemia) with three columns for control, ESCC samples, and p values?

6. As mentioned ML-based multiple-lipid models did not demonstrate markedly superior diagnostic performance compared to single-lipid models, how the model might be optimized/refined for larger samples or diverse patient groups?

7. The absence of this specific control group could potentially impact the interpretation of dysregulated lipid profiles and the development of the diagnostic model for ESCC?

8. Considering the promising results of the developed diagnostic model, what potential challenges and considerations are anticipated during the process of translating it into clinical practice?

9. Understanding the importance of validation, do the authors have plans to conduct further studies in diverse patient populations and clinical settings?

10. Compare the proposed diagnostic method using plasma-based lipidomics to existing diagnostic methods in terms of potential advantages, sensitivity, and specificity.

Annotated reviews are not available for download in order to protect the identity of reviewers who chose to remain anonymous.

---

## Round 0.2 · accepted · Accept

Based on the reports received from the reviewers I recommend the acceptance of the manuscript.

Reviewer 2 ·

Basic reporting

No comment

Experimental design

No comment

Validity of the findings

No comment

Additional comments

No comment

Reviewer 3 ·

Basic reporting

manuscript is well written, and the study's methodology and rationale are well-structured and scientifically informative

Experimental design

The experimental design is well executed

Validity of the findings

Conclusions read well and clearly with updated and supporting information